# Concurrent disease burden from multiple infectious diseases and the influence of social determinants in the contiguous United States

**Emma Blake, Este Stringham, Chantel Sloan-Aagard** [ID] *

Department of Public Health, Brigham Young University, Provo, UT, United States of America

* chantel.sloan@byu.edu

## Abstract

Social determinants of health are known to underly excessive burden from infectious diseases. However, it is unclear if social determinants are strong enough drivers to cause repeated infectious disease clusters in the same location. When infectious diseases are known to co-occur, such as in the co-occurrence of HIV and TB, it is also unknown how much social determinants of health can shift or intensify the co-occurrence. We collected available data on COVID-19, HIV, influenza, and TB by county in the United States from 2019–2022. We applied the Kulldorff scan statistic to examine the relative risk of each disease by year depending on the data available. Additional analyses using the percent of the county that is below the US poverty level as a covariate were conducted to examine how much clustering is associated with poverty levels. There were three counties identified at the centers of clusters in both the adjusted and unadjusted analysis. In the poverty-adjusted analysis, we found a general shift of infectious disease burden from urban to rural clusters.

## Introduction

Local transmission and prevalence of infectious diseases is highly dependent on a variety of factors including population density, vaccination coverage, social determinants of health, and human behavior relative to transmission mechanisms. [1, 2]. The impact of poverty and other deprivation metrics were exacerbated in certain geographic areas in the COVID-19 era [3–5]. While the underlying factors that create greater epidemic potential in certain neighborhoods are widely studied for individual conditions, cross-studies between multiple infectious diseases are rare [6–9].

Individuals living in areas with potential for multiple overlapping outbreaks are at higher risk of infection and co-infection. Previous research indicates that individuals co-infected with tuberculosis (TB), human immunodeficiency virus (HIV), and coronavirus disease (COVID-19) were at the highest risk of death and needed additional care during the COVID-19 era [10, 11]. Infection with TB worsened COVID-19 symptom severity and mortality rates and could be further worsened when co-infected with HIV [12–14]. Coinfection with influenza and COVID-19 may likewise increase mortality and worsen health outcomes [14], although the

**Data Availability Statement:** https://githubcom/nytimes/covid-19-data2021 https://www.cdc.gov/nchhstp/atlas/index.htm https://www.cdc.gov/flu/

weekly/overview.htm#Viral https://www.cdc.gov/
flu/fluvaxview/interactive.htm

**Funding:** The author(s) received no specific
funding for this work.

**Competing interests:** The authors have declared
that no competing interests exist.

influence varied depending on type. Some global geographic areas with high flu vaccination
rates were found to have lower mortality rates for COVID-19 [15].

Concurrent local epidemics can further impact the receipt of treatment and be an excessive
burden on health services. Services for TB and other diseases were decreased due to the
COVID-19 pandemic, and the disease burden increased due to the overwhelming rise in
COVID-19 cases in certain areas [2, 16, 17]. Investigating individual locations that have expe-
rienced past epidemics of multiple infectious diseases will help identify those that are at risk of
concurrent infections going forward and encourage additional local interventions that can
reduce overall disease burden. This study aims to examine concentrated multiple disease bur-
den in specific counties or geographic locales of the United States and determine which dis-
eases are more prevalent in these locations to identify patterns and assist efforts in combatting
infectious disease among the population.

## Methods

This study was a descriptive ecological study. It investigated infectious diseases from all contig-
uous counties in the US from the years 2019–2022 that data was available for the infectious dis-
eases, TB, HIV, COVID-19, and influenza. The data case counts were annual data, or data
collected from the total of each year.

### Data sources

Data for TB, HIV, COVID-19, and influenza were compiled from existing data sets.

(Data was collected from the seasons 2019–20 and 2020–21 and included all strains of influ-
enza) [18–20]. The data were organized by county and/or city subdivisions and we identified
the centroid coordinates of each location for spatial analyses. Data sets were chosen according
to the most recently collected data; the TB and HIV data was from years 2019 and 2020 while
the influenza data was from years 2020 and 2021, and the COVID-19 data was from years 2021
and 2022.

**COVID-19.**   The methods of reporting were different for each data set. COVID-19 data
were reported based on positive COVID-19 tests and compiled by the New York Times. The
surveillance case definition for COVID-19 [18] was the total number of both confirmed and
probable cases. Individuals with a confirmed case had a positive COVID-19 test from a labora-
tory test that was reported by a federal, state, territorial or local government agency. Only tests
that detected viral RNA in the sample were considered confirmatory. A probable case counted
individuals that did not have a confirmed test but were evaluated by public health officials
using criteria developed by states and the federal government and reported by a health depart-
ment. Laboratory, epidemiological, clinical and vital records were considered evidence by pub-
lic health officials. According to the Council of State and Territorial Epidemiologists, tests that
detected antigens or antibodies were considered evidence towards a probable case.

**HIV and TB.**   HIV and TB data were reported to the Centers for Disease Control and Pre-
vention (CDC) through local and state health departments and downloaded through the Atlas-
Plus portal. The surveillance case definition for TB [20, 21] was a case that met the clinical case
definition or was laboratory confirmed. The clinical criteria for a case diagnosis consisted of a
positive tuberculin skin test or positive interferon gamma release for *M. tuberculosis*, other
signs and symptoms compatible with TB (abnormal chest radiograph, abnormal chest com-
puterized tomography scan, etc.), treatment with two or more anti-TB medications, or a com-
pleted diagnostic evaluation. The laboratory criteria for a case diagnosis were the isolation of
*M. tuberculosis* from a clinical specimen, demonstration of *M. tuberculosis* complex from a
clinical specimen by nucleic acid amplification test, or the demonstration of acid-fast bacilli in

a clinical specimen when a culture has not been or cannot be obtained or is falsely negative or contaminated. The surveillance case definition for HIV [20, 22] was a confirmed case in one of the five HIV infection stages (0, 1, 2, 3, or unknown). If there was a negative HIV test within six months of the first HIV infection diagnosis, then the stage is zero. The other stages of HIV infection were determined by the age-specific CD4+ T-lymphocyte count or CD4+ T-lymphocyte percentage of total lymphocytes.

**Influenza.** Influenza data were reported by public health laboratories to formulate Flu-View's existing surveillance from the US WHO Collaborating Systems and NRVESS located throughout all 50 states of the United States [23]. For this study, data from the 48 contiguous states were used. The surveillance case definition for influenza [24] is a laboratory confirmation as well as signs and symptoms. A laboratory definition is virus isolation, molecular detection, detection of viral antigens or rapid influenza diagnostic tests, use of immunohistochemistry, or serologic testing using hemagglutination inhibition or microneutralization. The case definition used by the CDC in the US Outpatient Influenza-like Illness Surveillance Network, in which healthcare providers report the total number of patient visits and number of patients seen for ILI every week, is a fever 100°F or higher and a cough/sore throat. The influenza data was collected from the CDC Flu view for seasons 2019–20 and 2020–21 and includes all strains of influenza.

Different methods of data collection from various sources led to the possibility of inequalities in data comparison. Because of these potential differences, we chose to focus on overall patterns in disease distribution rather than individual cluster outputs.

## Statistical analysis

**Poisson model.** We employed a cluster detection method using the Kulldorff scan statistic implemented in the SaTScan software. We used the discrete Poisson probability model using case counts and underlying county populations to determine the ratios of observed to expected cases in a given year, identifying both high and low-risk areas. The maximal spatial cluster was set for 10% of the population at risk. The maximum number of replications was 999. Hierarchical priority was given to the most likely clusters and removed those of the same disease/year combination with geographical overlap. We ran a purely spatial analysis for each infectious disease for each year available, resulting in eight total unadjusted analyses (see S3 File).

A spatial cluster never contained more than 50% of the population at risk because a larger size is interpreted as a lower disease rate. The default maximum was 50%, but we chose a smaller maximum because it is difficult to make a meaningful interpretation of larger maximums. We originally set the maximum at 5%, but the maximum was too small for interpretation. We decided to set the maximum at 10% because it was small enough to be significant and large enough for interpretation.

The clusters were expressed in a circular window of kilometers. Large clusters indicated consistent higher disease burden across a geographical area. Smaller clusters indicated concentrations of high disease burden in smaller geographical areas or specific counties. The radius described in the results section was of high-risk clusters. The definition of a high-risk cluster for TB, HIV, COVID-19, and influenza was a geographical area with a significantly higher number of observed cases than expected cases. Groups of cases were defined as a high-risk cluster because of a pattern of high infectious disease burden in that geographical area.

Expected cases were determined by using a spatial scan statistic software [25] to analyze the geographical distribution of infectious disease cases in the contiguous United States. The spatial scan statistic evenly distributed the risk of each disease (i.e. TB cases for 2020) across counties, which was the expected case count. The expected cases were then compared to the

observed cases using the Poisson model to discover counties with a statistically significant disease burden.

**Poisson model with covariate.**   In this research, the dependent variable was the amount of a disease. It was measured by the number of observed cases in the data collected from the NY Times, US CDC influenza surveillance, and US CDC NCHHSTP. Location was the controlled independent variable. This research sought to find a relationship between location and disease burden. For example, the location of New York County had a significantly higher amount of COVID-19 and therefore individuals there were at greater risk than in another county. The controlled independent variable (location) affected the dependent variable (amount of disease).

Covariates are independent variables that can influence the outcome of a statistical trial. In this research, the covariate was the poverty level. The poverty levels of the percent of the population at or below 125% of the Federal Poverty Level were taken from the 2020 US Census for each county and used as an uncontrolled independent variable. Eight additional analyses (see S6 File) were completed by adjusting for underlying poverty levels. The covariate analysis was used to examine differences in the mean values of the dependent variables that are related to the effect of the controlled independent variable while taking into account the influence of the uncontrolled independent variable (percent below poverty level). It assumed a linear relationship between the dependent variable and the covariate. Therefore, by comparing the adjusted and unadjusted analyses, we can determine if there are strong patterns relative to socioeconomics that are driving cluster detection.

Results of the analyses frequently output single-county clusters. In order to compare across analyses, we focused on the central county of each cluster. Therefore, the results described refer to a single county, although it was not uncommon for clusters to span multiple counties. This study was not subject to IRB approval because we used freely available datasets. We were ethically careful not to display rates in counties with low numbers.

## Results

### Unadjusted analysis

The unadjusted analysis output 72 high risk clusters, or 72 areas where there were significantly higher cases of HIV, influenza, TB or COVID-19 in individual years than statistically expected, and 81 low risk clusters. We will focus mainly on the areas of high risk due to alignment with the study purpose. There were patterns of clusters that show a significantly higher amount of TB, HIV, and influenza in urban areas, the south, and along the coastline (see Fig 1). COVID-19 clusters were distributed differently and generally included a higher disease burden in areas that did not overlap with the other infectious diseases. For maps showing the central counties in each cluster, see S1 File.

Concentric HIV and TB high-risk clusters were identified as centered on Bronx County, NY, Kings County, NY, and Dallas County, TX (see S5 File). Significant high-risk clusters of COVID-19 and influenza were found in Clark County, NV, and COVID-19 and HIV clusters were found centered on Fairfield County, SC. The largest clusters were detected for COVID-19 in 2021 with a radius of 1045 miles, and COVID-19 in 2022 with a radius of 906 miles in the North-Central part of the US. Despite being the largest in size, these clusters did not overlap with other infectious disease high risk areas. Most of the other diseases had some overlap.

### Poverty-adjusted analysis

The poverty-adjusted analysis output 280 high risk clusters (see Fig 2). All of the counties in the poverty-adjusted analysis that were centers of high-risk clusters for COVID-19 in

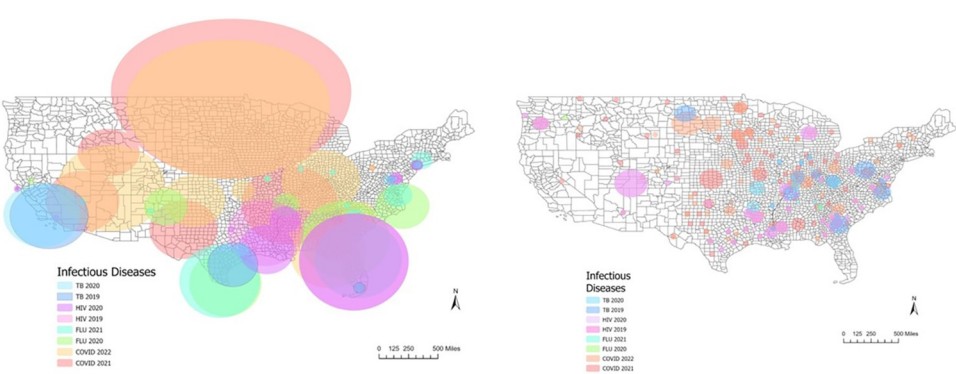

**Fig 1. High-risk cluster locations in the unadjusted analysis and poverty-adjusted analysis.** This figure shows high-risk cluster locations and radii for disease and year for the unadjusted analysis (left), and high-risk cluster locations and radii for disease and year for the poverty-adjusted analysis (right). The shapefile used was from the US Census Bureau publicly available TIGER/Line Shapefiles. Counties at the center of high-risk clusters in consecutive years (see S4 File) were found for COVID-19, HIV, and Influenza. Central counties for high-risk clusters for COVID-19 were in Miami-Dade County, FL, New York County, NY, Westmoreland County, PA and Yuma County, AZ. Counties at the center of regions with high disease burden of HIV in consecutive years were Anne Arundel County, MD, Lake County, FL, Los Angeles County, CA and San Francisco County, CA. The county with a high disease burden of influenza in consecutive years was Boone County, MO. Based on this analysis, none of the counties were at the center of high-risk clusters for TB in consecutive years.

consecutive years had populations of less than 100,000 people. These included counties in TN, MN, TX, KY, CO, MO, MN, WI, VA, IL, and NC. When investigating rates of HIV in consecutive years, all counties were found to be at the center of a high-risk cluster and had

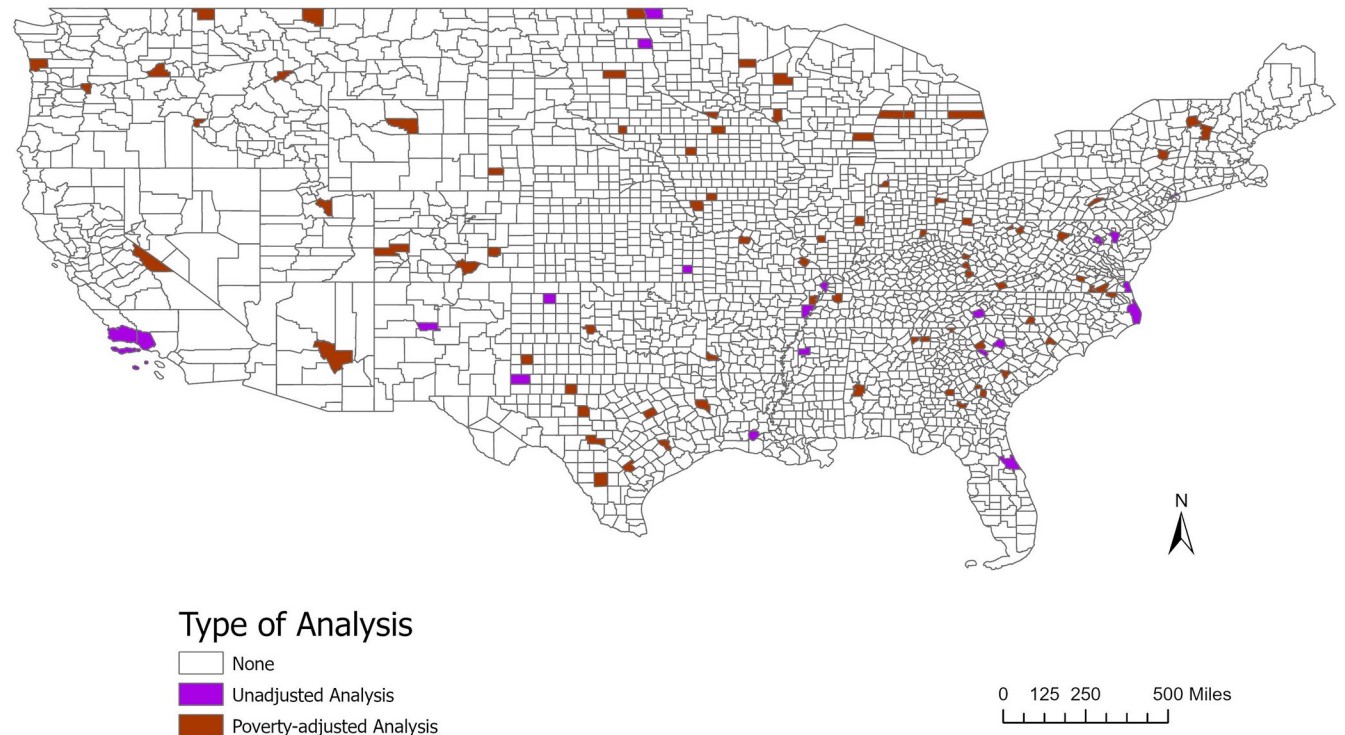

**Fig 2. Analysis type for counties found in high-risk clusters.** This figure shows the counties that were found to be at the center of a high relative risk cluster by analysis type: unadjusted or poverty-adjusted (adjusted for 125% below the poverty level). The shapefile used was from the US Census Bureau publicly available TIGER/Line Shapefiles.

populations less than 100,000. There was only one county that was at the center of an influenza cluster in consecutive years, Gordon County, GA (see S7 File). There were two counties identified as cluster centers for TB in consecutive years, Fulton County, IN and Richmond County, VA.

There was a spatial correlation between COVID-19 for years 2021 and 2022 and HIV for years 2019 and 2020 (see S8 File). Clusters were found with overlaps in 16 counties primarily in the south and southeast US. This included 11 counties that were centers of high-risk clusters for HIV in consecutive years 2019 and 2020 (in addition to a high-risk cluster of COVID-19 in either 2021or 2022), and three that were centers of high-risk clusters for COVID-19 in consecutive years 2021 and 2022 (in addition to a high-risk cluster of HIV in either 2019 or 2020). There were two counties, Kerr County, TX and Stanly County, NC, that had high-risk clusters for HIV in 2019 and 2020 and high-risk clusters for COVID-19 in 2021 and 2022.

There were two counties with clusters of COVID-19 and influenza (see S8 File), Boyle County, KY and Gordon County, GA. Overlap between three or more disease cluster types was rare, however. Richmond County, VA had overlapping clusters of COVID-19, HIV, and TB. In the poverty-adjusted analysis, the largest clusters were for COVID-19 (2022) with a radius of 129 miles, and COVID-19 (2021) with a radius of 104 miles.

The percentage of the county population at or below the US poverty level in counties identified as part of clusters ranged from 7.65% to 44.8%. Most of the clusters were centered on counties central to high-risk clusters had between 10 and 29.9% of their population below the poverty line (see S2 File).

**Patterns across analyses.** The type of analysis used in relation to the counties found in the center of high-risk clusters was examined (Fig 3). There were three counties identified at the centers of one or more disease clusters in both of the analyses. These counties were Grady County, GA, Hansford County, TX, and Scott County, IN. The percent of the population below the poverty line was 23.43%, 26.28%, and 24.19% respectively. All of these counties had high-risk clusters for COVID-19. Each of these three counties were found in rural areas. Grady County, GA and Scott County, IN had populations of around 25,000 people, while Hansford County had a population of about 5,000 people.

The overlap of clusters indicated that several counties were at a high relative risk for disease burden and concurrent disease burden. An analysis of the counties at the center of disease clusters and the type of associated infectious disease was completed (Fig 4). These counties were found to be at a higher risk for one or multiple diseases.

There was an increase in the number of clusters from the unadjusted analysis (see Fig 5) to the poverty-adjusted analysis except for influenza. There were about eight times as many COVID-19 clusters, four times as many HIV clusters, and two or three times as many TB clusters in the poverty-adjusted analysis. There were a quarter as many influenza clusters. It was expected that the poverty-adjusted analysis would decrease the number of clusters and the higher infectious disease burden would be explained by poverty. Instead, there was an increase of clusters with smaller radii in rural areas, indicating that rural poverty is a driver of infectious disease burden.

## Discussion

The unadjusted analysis showed higher numbers of clusters of TB, HIV, and influenza in urban areas, the south, and along the coastline, while COVID-19 clusters appeared separately. Upon adjusting for poverty, more localized and rural clusters were found in the interior of the country for HIV, TB and COVID-19.

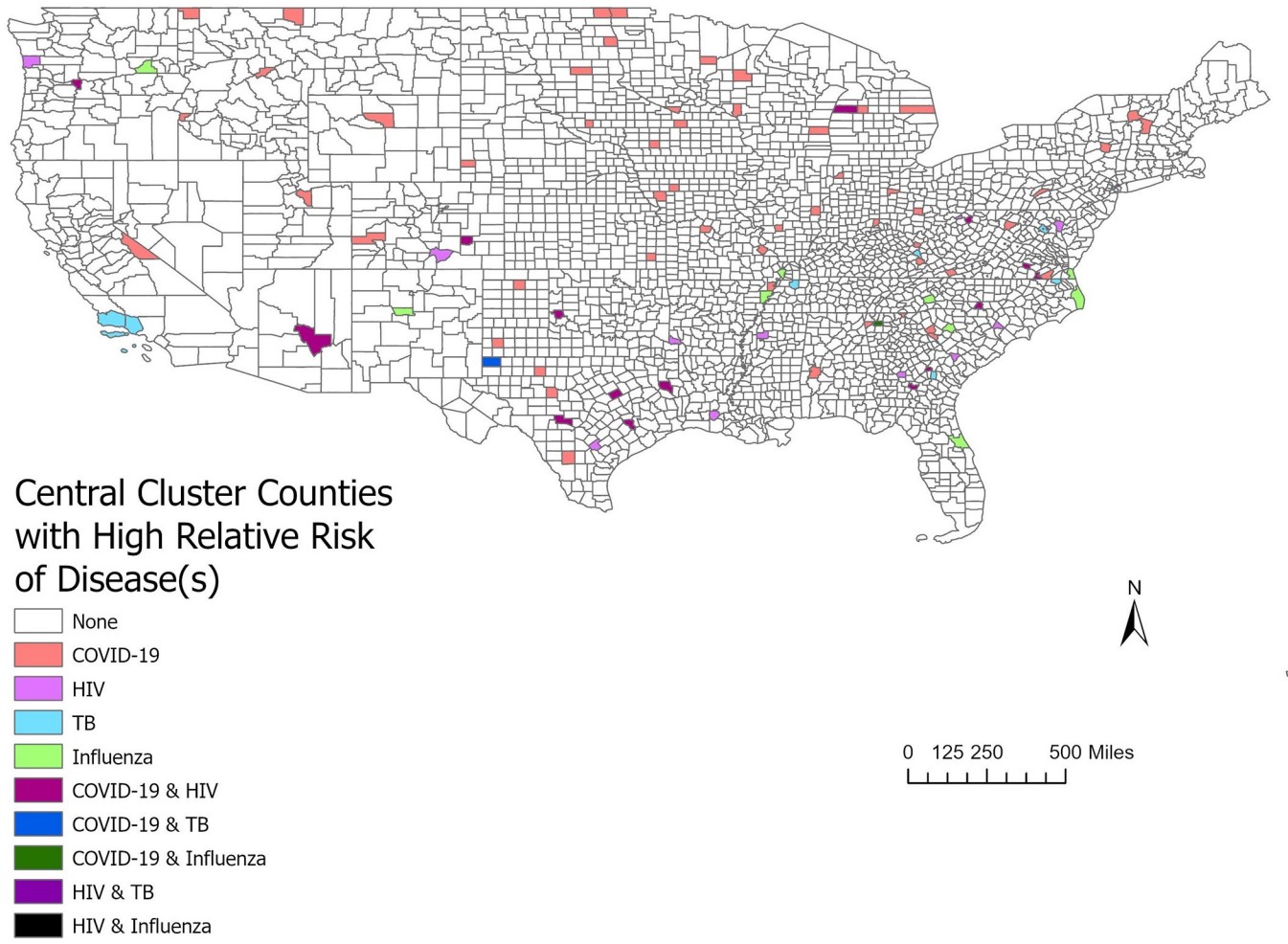

**Fig 3. Counties found in high-risk clusters with one or multiple diseases.** This figure shows the counties that were at the center of clusters by disease, one or multiple diseases. The shapefile used was from the US Census Bureau publicly available TIGER/Line Shapefiles.

High population density in urban areas is known to be associated with infectious disease spread [26, 27]. On the other hand, rural areas face infrastructure and staffing challenges, making it difficult to prevent and respond to infectious disease outbreaks [27]. The distribution of infectious diseases in rural vs urban areas in the unadjusted analysis was found to be different depending on the infectious disease. A high relative risk for HIV was consistently more prevalent in urban areas. This could be attributed to certain lifestyles that are more common or accepted in urban settings.

Influenza clusters were few, which was expected given the low counts of influenza during the COVID-19 era [28]. What clusters were found, persisted specifically in New Mexico and neighboring regions in the southwestern US. There is a possible vaccination gap in this area of the US, although more research would need to be conducted to see if high rates of influenza are correlated with influenza vaccination coverage in southwest states [29]. Both TB and COVID-19 had clusters with high relative risks for both urban and rural areas. Due to the COVID-19 pandemic, TB case diagnoses may have been missed. Data from 2021 to 2022 indicates that TB cases increased but were still lower than they were in 2019 [30].

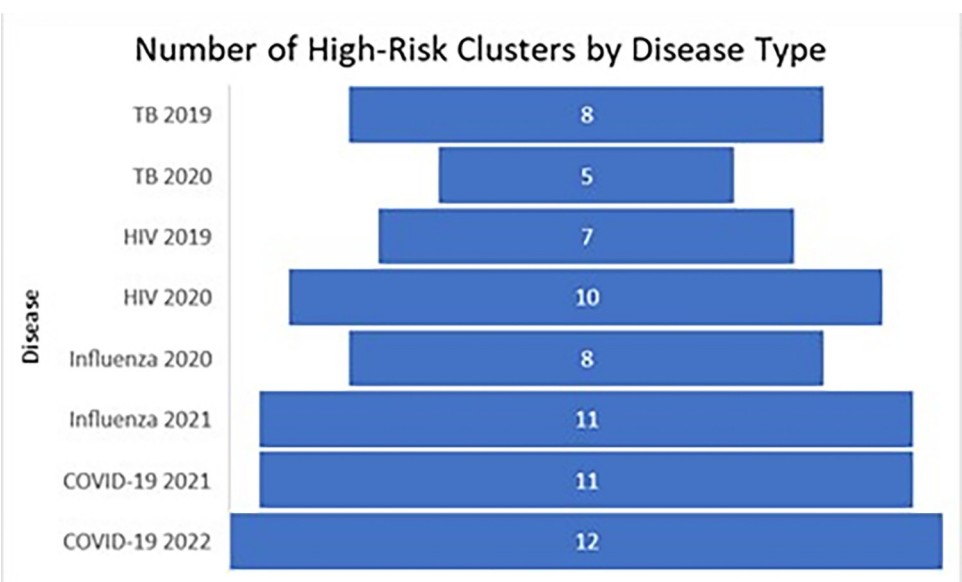

**Fig 4. High-risk clusters attributed to type of infectious disease in the unadjusted analysis.** These figures show the number of high-risk clusters attributed to the type of infectious disease from the unadjusted analysis.

In the US, urban areas experienced a higher incidence rate of COVID-19 for the first several months of the pandemic, and then the higher rate of incidents shifted over to rural areas [31]. The spread of infectious disease in rural areas was possibly due to localization, or increased exposure at locations, such as a gas station, where individuals go to obtain needed supplies. Rural areas were also at a high risk of morbidity and mortality for COVID-19 because of the combined factors of high disease burden and low healthcare capacity [31].

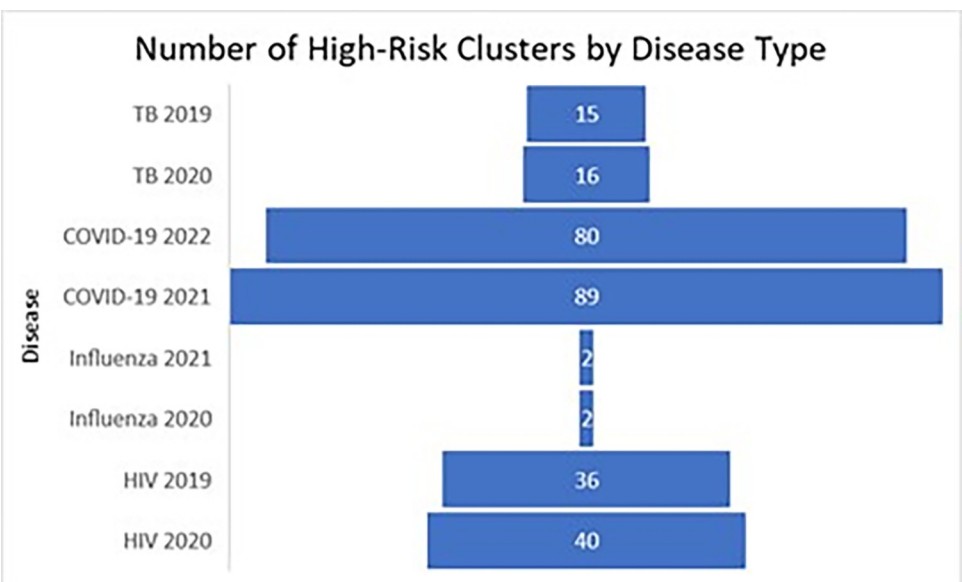

**Fig 5. High-risk clusters attributed to type of infectious disease in the poverty-adjusted analysis.** These figures show the number of high-risk clusters attributed to the type of infectious disease from the poverty-adjusted analysis.

When adjusting for poverty, the most striking overall pattern was a shift from large clusters primarily centered on urban areas to many small clusters in more rural areas in the interior of the country. This suggests that the major driver of statistically high rates of infectious disease in the United States is (or is closely correlated with) poverty. In past studies of influenza clustering, poverty adjustment decreased the overall number of clusters by almost 90% [32]. The pattern of reduction for influenza persisted in this analysis, however the results were very different for COVID-19, HIV and TB. Here we saw a proliferation of clusters. After removing the statistical influence of poverty, there may be more localized effects associated with rural areas including lack of access to medical services, lower vaccination coverage, etc. that drives variability in disease rates. Those influences for COVID-19 are especially interesting and worthy of investigation, including whether clusters persisted in the same locations during different types of COVID-19 (wild type, delta, omicron).

The study was limited in that we were relying on separate data sources for each infectious disease, with different collection methods, accuracy, and some different years of availability. This is why we focused primarily on overall changes in patterns of disease rather than focusing on identifying specific counties for interventions. However, we do present county data in the supporting information if others are interested in comparing our results with their own.

## Conclusion

Poverty is associated with the patterns of disease clusters for multiple infectious diseases, including HIV, TB, COVID-19 and influenza. However, following adjustment for poverty the clusters occurred in different locations and in different patterns, though mostly moving from large urban-centric clusters to smaller rural-centric clusters. This suggests that localized variation relative to mechanisms of disease spread play strong secondary roles for COVID-19, HIV and TB, and that variations leading to higher disease burden tend to be more pronounced in rural areas.

In both urban and rural areas, poverty is associated with statistically high rates of infectious disease in the United States. It is important to consider the impact that health interventions have on each of the urban and rural populations. Health interventions should be adjusted to meet the needs of the population to improve the social determinants of health in that community and combat poverty in the United States.

## Supporting information

**S1 File. File containing a map of the central cluster counties with the highest relative risk for each disease in the study, a second map showing the central cluster counties for multiple diseases, and a graph of the number of high-risk clusters by percent of the population below the poverty line.**
(DOCX)

**S2 File. File with tables containing the data supporting figures found in S1 File.**
(DOCX)

**S3 File. This file contains several tables showing the results for the unadjusted clusters for COVID-19, HIV, Influenza, and TB in the years 2019–2022.** Included in the tables are the county name, state, p-value, expected number of cases, observed number of cases, the relative risk for the disease, and the county population.
(DOCX)

**S4 File. File containing tables which list counties with a high relative risk for the same disease in consecutive years.** Included in the table are the county name, state, p-value, expected

number of cases, observed number of cases, the relative risk for the disease, and the county population.
(DOCX)

**S5 File. This file contains tables listing the counties that had a high relative risk for two different diseases in the unadjusted analysis.** Included in the table are the county name, state, p-value, expected number of cases, observed number of cases, the relative risk for the disease, and the county population.
(DOCX)

**S6 File. The included tables list the counties that were at a high relative risk for each of the diseases, COVID-19, HIV, Influenza, and TB in the years 2019–2022 in the adjusted analysis.** Included in the table are the county name, state, p-value, expected number of cases, observed number of cases, the relative risk for the disease, the county population, the number of individuals below the poverty line in the county, and the percent of the county population that is 125% below the US poverty line.
(DOCX)

**S7 File. The tables included in this file list the counties with a high relative risk for the same disease in consecutive years, adjusted for poverty.** Included in the table are the county name, state, p-value, expected number of cases, observed number of cases, the relative risk for the disease, the county population, the number of individuals below the poverty line in the county, and the percent of the county population that is 125% below the US poverty line.
(DOCX)

**S8 File. File containing tables which show the counties that had a high relative risk for two different diseases, adjusted for poverty.** Included in the table are the county name, state, p-value, expected number of cases, observed number of cases, the relative risk for the disease, the county population, the number of individuals below the poverty line in the county, and the percent of the county population that is 125% below the US poverty line.
(DOCX)

## Author Contributions

**Conceptualization:** Chantel Sloan-Aagard.

**Data curation:** Emma Blake.

**Formal analysis:** Emma Blake.

**Methodology:** Chantel Sloan-Aagard.

**Supervision:** Chantel Sloan-Aagard.

**Visualization:** Emma Blake.

**Writing – original draft:** Emma Blake, Este Stringham.

**Writing – review & editing:** Emma Blake, Este Stringham.

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
