## [Decision Letter · Decision Letter 0]

31 Jul 2023

PONE-D-23-15168Concurrent Disease Burden from Multiple Infectious Diseases and the Influence of Social Determinants in the Contiguous United StatesPLOS ONE

Dear Dr. Sloan-Aagard,

Thank you for submitting your manuscript to PLOS ONE. After careful consideration, we feel that it has merit but does not fully meet PLOS ONE’s publication criteria as it currently stands. Therefore, we invite you to submit a revised version of the manuscript that addresses the points raised during the review process.

ACADEMIC EDITOR: 

The manuscript addresses an important issue and has merit. However, authors must carefully attend to the concerns raised by the reviewers. This involves addressing various aspects:

Ensuring language and terminology are clear in the entire text, tables, and figures.Expanding on the study's justification and objectives for better understanding.Providing necessary clarifications on study design, datasets, definitions, and analyses.Enhancing graph color contrast for better visual clarity.Elaborating further on study conclusions and their implications.

By addressing these points, the authors can substantially enhance the manuscript's clarity, validity, and impact.

We look forward to receiving your revised manuscript.

Kind regards,

Sana Eybpoosh

Academic Editor

PLOS ONE

2. We note that Figures 1A,1B,2 and 3 in your submission contain [map/satellite] images which may be copyrighted. All PLOS content is published under the Creative Commons Attribution License (CC BY 4.0), which means that the manuscript, images, and Supporting Information files will be freely available online, and any third party is permitted to access, download, copy, distribute, and use these materials in any way, even commercially, with proper attribution. For these reasons, we cannot publish previously copyrighted maps or satellite images created using proprietary data, such as Google software (Google Maps, Street View, and Earth). For more information, see our copyright guidelines: http://journals.plos.org/plosone/s/licenses-and-copyright.

a. You may seek permission from the original copyright holder of Figures 1A,1B,2 and 3  to publish the content specifically under the CC BY 4.0 license. 

Reviewers' comments:

Reviewer's Responses to Questions

**Comments to the Author**

1. Is the manuscript technically sound, and do the data support the conclusions?

Reviewer #1: Yes

Reviewer #2: Yes

2. Has the statistical analysis been performed appropriately and rigorously? 

Reviewer #1: I Don't Know

Reviewer #2: Yes

3. Have the authors made all data underlying the findings in their manuscript fully available?

Reviewer #1: Yes

Reviewer #2: Yes

4. Is the manuscript presented in an intelligible fashion and written in standard English?

Reviewer #1: Yes

Reviewer #2: Yes

5. Review Comments to the Author

Reviewer #1: Abstract:

Please write numbers less than 10 in letters (in the whole of the text)

Introduction:

Please replace "vaccination rates" with vaccination coverage.

Please justify the necessity of this study and write the aim of the study, in the last paragraph of the introduction.

Methods:

What is your study design?

What are the existing data sets? Do you mean the surveillance system?

Please refer to the source of data. In addition, please provide the definition of TB, HIV, COVID-19, and influenza in the US surveillance system.

Please provide the definition of a cluster for TB, HIV, COVID-19, and Influenza.

How to determine the expected cases?

What is the definition of high-risk and low-risk areas and clusters?

Please define the dependent variable. How to measure the dependent variable?

What are the covariates, and how to measure them?

How to assess the socioeconomic status of counties?

Results

In the first paragraph of the results, describe your data. For example, number of counties and incidence of the mentioned diseases in each county, and percent of the population below the poverty line in each county.

The authors declared "The percent of the county population at or below 125% of the US poverty level in counties identified as part of clusters ranged from 7.65% to 44.8%" 125% is correct?

Reviewer #2: Comments of Reviewer

This study aims to assess concurrent disease burden from multiple infectious diseases and the influence of social determinants in the contiguous United States. They applied the Kulldorff scan statistic to examine the relative risk of each disease by year depending on available data on COVID-19, HIV, influenza, and tuberculosis by county in the United States from 2019-2022. The unadjusted analysis showed higher numbers of clusters of TB, HIV, and influenza in urban areas, the south, and along the coastline, while COVID-19 clusters appeared separate. Upon adjusting for poverty, much more localized and rural clusters more in the interior of the country for HIV, TB and COVID-19. In my opinion, the obtained results can be used to inform policy makers and the public health. However, there are some issues that need to be addressed.

Major

1- I have concerns about the structure of this manuscript. The introduction and method of this manuscript is written very briefly. This manuscript does not have a conclusion section; it seems that it is better to add conclusion to the manuscript.

Abstract

Introduction

1- The introduction section is written very briefly, it is better to explain more about a brief statement of the overall aim of the work and a comment about whether that aim was achieved in the last paragraph. Moreover, please define the problem addressed and why it is important.

Method

1- Please clarify that you used the radius of the population coverage or the geographical radius? Why?

2- Please clarify that the discrete Poisson probability model or Poisson model was used because the incidence for some disease like TB and HIV were not very high

3- Please explain more about the most likely cluster area, and other clusters.

4- Please explain more about the selection of the maximum radius of the spatial

scanning window and the maximum length of the temporal scanning window. Why did you consider the maximal spatial cluster 10% of the population at risk?

5- Please clarify that you used monthly or annual incidences.

6- Please explain more about circular window.

7- Please clarify that did your study have an ethical approval and ethical code?

Results

1- Please move the first paragraph of the results in to the method section.

2- Please change the colors in the Graphs, the colors are not recognizable.

6. PLOS authors have the option to publish the peer review history of their article (what does this mean?). If published, this will include your full peer review and any attached files.

Reviewer #1: No

Reviewer #2: No

---

## [Author Response · Author response to Decision Letter 0]

14 Sep 2023

Responses to Editor and Reviewers

A. Comments from the Editor:

1. Editor: Please ensure that your manuscript meets PLOS ONE's style requirements, including those for file naming. The PLOS ONE style templates can be found at

Response: Thank you for reminding us of the templates. We reviewed the manuscript and changed the heading font size to 18 for level 1 headings, to font size 16 for level 2 headings, and to font size 14 for level 3 headings. We added page and line numbers to the document and inserted paragraph indentations. We changed figure citations to “Fig 1A” instead of “Figure 1A,” and capitalized table citations from “tables 1-3” to “Tables 1-3.” We revised the captions for Figures 1A, 1B, 2, 3, 4A and 4B to list the figure name, title, and legend in the format shown in the template. The tables are lengthy and were maintained in the supplement to the manuscript. The files for the figures were renamed “Fig 1A.tif” etc. to fit the requirements. 

We adjusted the title so that the first word of the title and proper nouns were the only capitalized words. We examined the names and affiliations to ensure that it was accurate and according to the guidelines.

2. Editor: We note that Figures 1A,1B,2 and 3 in your submission contain [map/satellite] images which may be copyrighted. All PLOS content is published under the Creative Commons Attribution License (CC BY 4.0), which means that the manuscript, images, and Supporting Information files will be freely available online, and any third party is permitted to access, download, copy, distribute, and use these materials in any way, even commercially, with proper attribution. For these reasons, we cannot publish previously copyrighted maps or satellite images created using proprietary data, such as Google software (Google Maps, Street View, and Earth). For more information, see our copyright guidelines: http://journals.plos.org/plosone/s/licenses-and-copyright.

Response: The images were created using tiger shapefiles (https://www.census.gov/geographies/mapping-files/time-series/geo/tiger-line-file.html) and data from the cited data sources (NY Times, CDC, and US Influenza Surveillance) into the software ArcGIS Pro. On page 9 of the shapefile technical documentation, it states, “Copyright protection is not available for any work of the United States Government (Title 17 U.S.C., Section 105). Thus, you are free to reproduce census materials as you see fit. We would ask, however, that you cite the Census Bureau as the source.” The images in this manuscript are not previously copyrighted and are original to the research. We added citations for the US Census Bureau for the map shapefiles throughout the manuscript.

3. Editor: Please include captions for your Supporting Information files at the end of your manuscript, and update any in-text citations to match accordingly. Please see our Supporting Information guidelines for more information.

Response: Thank you for explaining that the captions for the supporting information should be at the end of the manuscript. After the references, we added the titles for the captions in the format described. Many of the captions have legends included. 

B. Comments from Reviewer #1: 

1. Reviewer: 

Abstract: 

Please write numbers less than 10 in letters (in the whole of the text)

Response: Thank you for your correction. The numbers less than 10 were written out in letters. On line 15, “3” was changed to “three”. On line 127, “3” was changed to “three”. 

2. Reviewer: 

Introduction:

Please replace "vaccination rates" with vaccination coverage.

Please justify the necessity of this study and write the aim of the study, in the last paragraph of the introduction.

Response: We replaced "Vaccination rates" was replaced with vaccination coverage. 

The following sentence was added to the end of the introduction to justify and clarify the aim of the study, “This study aims to examine whether counties in the United States experience concurrent excessive burden from multiple infectious diseases and identify if those patterns are associated with underlying socioeconomic status.”

3. Reviewer: The reviewer provided an excellent series of questions related to the Methods Section. We answer each in turn below and in the document.

What is your study design?

This study was a descriptive ecological study. It investigated infectious diseases across counties in the US at a point in time from the years for which data were available. 

What are the existing data sets? Do you mean the surveillance system? Please refer to the source of data.

The existing data sets were taken from the NY Times, US CDC influenza surveillance, and US CDC NCHHSTP as described in the manuscript, found in references, 18, 19 and 20 respectively. 

18. Times TNY. Coronavirus (Covid-19) Data in the United States.

19. Centers for Disease Control and Prevention. US influenza surveillance: purpose and methods. US Influenza Surveillance: Purpose and Methods CDC. 2021.

20. Centers for Disease Control and Prevention. NCHHSTP AtlasPlus 2023.

In addition, please provide the definition of TB, HIV, COVID-19, and influenza in the US surveillance system.

The surveillance case definition for COVID-19 provided by the NY times is the total number of both confirmed and probable cases. Individuals with a confirmed case had a positive laboratory COVID-19 test reported by a federal, state, territorial or local government agency. Only tests that detected viral RNA in the sample were considered confirmatory. A probable case counted individuals that did not have a confirmed test but were evaluated by public health officials using criteria developed by states and the federal government and reported by a health department. Laboratory, epidemiological, clinical and vital records were considered evidence by public health officials. Tests that detected antigens or antibodies were considered evidence towards a “probable” case, but were not sufficient on their own, according to the Council of State and Territorial Epidemiologists. 

The surveillance case definition for TB was a case that met the clinical case definition or was laboratory confirmed. The clinical case definition included a positive tuberculin skin test or positive interferon gamma release for M. tuberculosis, other signs and symptoms compatible with TB such as an abnormal chest radiograph or abnormal chest computerized tomography scan, treatment with two or more anti-TB medications, or a completed diagnostic evaluation. The laboratory criteria for a case diagnosis was isolation of M. tuberculosis from a clinical specimen, demonstration of M. tuberculosis complex from a clinical specimen by nucleic acid amplification test, or the demonstration of acid-fast bacilli in a clinical specimen when a culture has not been or cannot be obtained or is falsely negative or contaminated.

The surveillance case definition for influenza was laboratory confirmation as well as signs and symptoms. A laboratory definition included virus isolation, molecular detection, detection of viral antigens or rapid influenza diagnostic tests, use of immunohistochemistry, or a serologic testing using hemagglutination inhibition or microneutralization. The case definition was a fever 100⁰F or higher and a cough/sore throat used by the CDC in the US Outpatient Influenza-like Illness Surveillance Network, in which healthcare providers report the total number of patient visits and number of patients seen for ILI every week. The influenza data was collected from the CDC Flu view for seasons 2019-20 and 2020-21 and includes all influenza strains.

The surveillance case definition for HIV was a confirmed case in one of the five HIV infection stages (0, 1, 2, 3, or unknown). After the first HIV infection diagnosis, if there was a negative HIV infection diagnosis then the stage is 0. The other stages are determined by the following table:

HIV infection stage, based on age-specific CD4+ T-lymphocyte count or CD4+ T-lymphocyte percentage of total lymphocytes

Stage Cells (<1 year) % (<1 year) Cells (1-5 years) % (1-5 years) Cells (6 years) % (6 years)

1 ≥1500 ≥34 ≥1000 ≥30 ≥500 ≥26

2 750-1499 26-33 500-999 22-29 200-499 14-25

3 <750 <26 <500 <22 <200 <14

The above information and table was made available from the following source: Centers for Disease Control and Prevention. Terms, Definitions, and Calculations Surveillance Overview HIV 2022 [Available from: https://www.cdc.gov/hiv/statistics/surveillance/terms.html

Please provide the definition of a cluster for TB, HIV, COVID-19, and Influenza.

The definition of a cluster for TB, HIV, COVID-19, and influenza was a geographical area with a significantly higher number of observed cases than expected cases. 

How to determine the expected cases? What is the definition of high-risk and low-risk areas and clusters?

Expected cases were determined using a spatial scan statistic software (SaTScan) to analyze the geographical distribution of infectious disease cases in the contiguous United States. The spatial scan statistic evenly distributed the risk of each disease (i.e. TB cases for 2020) across counties, which is the expected case count. The expected cases were then compared to the observed cases using a Markov Chain Monte Carlo method to discern counties with a statistically higher or lower than expected cases according to likelihood. Counties with an associated p-value <0.05 were considered significant. We refer the reader to the SaTScan documentation for more detail. 

Please define the dependent variable. How to measure the dependent variable?

The dependent variable was frequency of occurrence of each disease in each county within a given year, adjusted by the underlying population. 

What are the covariates, and how to measure them?

The primary covariate was the percent of the population at or below 125% of the Federal poverty level as obtained from the 2020 US Census, as described. 

How to assess the socioeconomic status of counties?

Socioeconomic status can be measured using multiple variables. We selected the percentage of the population at or below 125% of the Federal Poverty Level because poverty is associated with many health-related outcomes in the US, including access to care and insurance coverage. 

4. Reviewer: In the first paragraph of the results, describe your data. For example, number of counties and incidence of the mentioned diseases in each county, and percent of the population below the poverty line in each county.

The authors declared "The percent of the county population at or below 125% of the US poverty level in counties identified as part of clusters ranged from 7.65% to 44.8%" 125% is correct?

Response: Thank you for the feedback. We included more information describing the data in the first paragraph. Yes, the statement is accurate according to the research, but we rephrased it for clarification. 

C. Comments from Reviewer #2:

1. Reviewer: I have concerns about the structure of this manuscript. The introduction and method of this manuscript is written very briefly. This manuscript does not have a conclusion section; it seems that it is better to add conclusion to the manuscript.

Response: Thank you for your feedback. Additions were included in the introduction and methods sections to better explain the process of this research. A conclusion was added to the manuscript.

2. Reviewer: 

Introduction

The introduction section is written very briefly; it is better to explain more about a brief statement of the overall aim of the work and a comment about whether that aim was achieved in the last paragraph. Moreover, please define the problem addressed and why it is important.

Response: Thank you, we agree that more information should be added to the introduction. We added, “This study aims to examine whether counties in the United States experience concurrent excessive burden from multiple infectious diseases and identify if those patterns are associated with underlying socioeconomic status.”

3. Reviewer: The reviewer provided a series of excellent comments related to the methods section. We answer each in turn below, and corresponding updated language can be found in the manuscript. 

Please clarify that you used the radius of the population coverage or the geographical radius? Why?

The radius described in the manuscript was of each high-risk cluster. This cluster indicates the size of the geographic area with a significantly high risk of a specific infectious disease, and is standard SaTScan output. 

Please clarify that the discrete Poisson probability model or Poisson model was used because the incidence for some disease like TB and HIV were not very high.

The discrete Poisson probability model was used in SaTScan. It was a purely spatial analysis. Incidence for TB cases were lower than the other diseases, and this is because of lower observation rates in counties throughout the US. The discrete Poisson probability model is better for low case counts.

Please explain more about the most likely cluster area, and other clusters.

High-risk clusters are geographic areas with a statistically significant concentration of one or multiple infectious diseases in the county. The most likely cluster is the high-risk cluster with the highest maximum likelihood. We report this as well as other clusters with a p-value <0.05. 

Please explain more about the selection of the maximum radius of the spatial scanning window and the maximum length of the temporal scanning window. Why did you consider the maximal spatial cluster 10% of the population at risk?

The default maximum is 50% of the population, but we chose a smaller maximum because larger maximums are difficult to make a meaningful interpretation of. The choice of 10% followed some experimentation with different size radii, and seemed a good balance of being able to identify rural and urban clusters that improved our understanding of national patterns (not very tiny or very large). 

Please clarify that you used monthly or annual incidences.

We used annual incidence. 

Please explain more about circular window.

The SaTScan standard is to create circular windows, though elliptical windows are an option. We find elliptical windows to be less informative because you can have many overlapping ellipses rotated on different axis. 

Please clarify that did your study have an ethical approval and ethical code?

This study was not subject to IRB approval because we used freely available datasets. However, we were very careful to use ethical mapping principles, including not displaying rates in counties with very low case counts. 

4. Reviewer:

Please move the first paragraph of the results into the method section.

Please change the colors in the Graphs, the colors are not recognizable.

Response: Thank you. The first paragraph of the results section was moved to the end of the methods section. 

We considered changing the colors of the graphs to make it clear which disease was which, but it was distracting, and we determined to keep the colors the same color as before in order to maintain the focus on the comparison of the number high-risk clusters and disease type.

---

## [Decision Letter · Decision Letter 1]

12 Oct 2023

Concurrent Disease Burden from Multiple Infectious Diseases and the Influence of Social Determinants in the Contiguous United States

PONE-D-23-15168R1

Dear Dr. Chantel Sloan-Aagard,

We’re pleased to inform you that your manuscript has been judged scientifically suitable for publication and will be formally accepted for publication once it meets all outstanding technical requirements.

Kind regards,

Sana Eybpoosh

Academic Editor

PLOS ONE

Additional Editor Comments (optional):

Reviewers' comments:

Reviewer's Responses to Questions

**Comments to the Author**

1. If the authors have adequately addressed your comments raised in a previous round of review and you feel that this manuscript is now acceptable for publication, you may indicate that here to bypass the “Comments to the Author” section, enter your conflict of interest statement in the “Confidential to Editor” section, and submit your "Accept" recommendation.

Reviewer #2: All comments have been addressed

Reviewer #3: All comments have been addressed

2. Is the manuscript technically sound, and do the data support the conclusions?

Reviewer #2: Yes

Reviewer #3: Yes

3. Has the statistical analysis been performed appropriately and rigorously? 

Reviewer #2: Yes

Reviewer #3: I Don't Know

4. Have the authors made all data underlying the findings in their manuscript fully available?

Reviewer #2: Yes

Reviewer #3: Yes

5. Is the manuscript presented in an intelligible fashion and written in standard English?

Reviewer #2: Yes

Reviewer #3: Yes

6. Review Comments to the Author

Reviewer #2: (No Response)

Reviewer #3: (No Response)

7. PLOS authors have the option to publish the peer review history of their article (what does this mean?). If published, this will include your full peer review and any attached files.

Reviewer #2: No

Reviewer #3: **Yes: **Amin Doosti-Irani
